# Unraveling a Force-Generating Allosteric Pathway of Actomyosin Communication Associated with ADP and P_i_ Release

**DOI:** 10.3390/ijms22010104

**Published:** 2020-12-24

**Authors:** Peter Franz, Wiebke Ewert, Matthias Preller, Georgios Tsiavaliaris

**Affiliations:** 1Cellular Biophysics, Institute for Biophysical Chemistry, Hannover Medical School, 30625 Hannover, Germany; franz.peter@mh-hannover.de; 2Structural Bioinformatics and Chemical Biology, Institute for Biophysical Chemistry, Hannover Medical School, 30625 Hannover, Germany; ewert.wiebke@mh-hannover.de (W.E.); matthias.preller@h-brs.de (M.P.); 3Department of Natural Sciences, University of Applied Sciences Bonn-Rhein-Sieg, 53757 Sankt Augustin, Germany

**Keywords:** myosin, actin, ATPase cycle, transient kinetics, duty ratio, allosteric communication, force generation, power stroke

## Abstract

The actomyosin system generates mechanical work with the execution of the power stroke, an ATP-driven, two-step rotational swing of the myosin-neck that occurs post ATP hydrolysis during the transition from weakly to strongly actin-bound myosin states concomitant with P_i_ release and prior to ADP dissociation. The activating role of actin on product release and force generation is well documented; however, the communication paths associated with weak-to-strong transitions are poorly characterized. With the aid of mutant analyses based on kinetic investigations and simulations, we identified the W-helix as an important hub coupling the structural changes of switch elements during ATP hydrolysis to temporally controlled interactions with actin that are passed to the central transducer and converter. Disturbing the W-helix/transducer pathway increased actin-activated ATP turnover and reduced motor performance as a consequence of prolonged duration of the strongly actin-attached states. Actin-triggered P_i_ release was accelerated, while ADP release considerably decelerated, both limiting maximum ATPase, thus transforming myosin-2 into a high-duty-ratio motor. This kinetic signature of the mutant allowed us to define the fractional occupancies of intermediate states during the ATPase cycle providing evidence that myosin populates a cleft-closure state of strong actin interaction during the weak-to-strong transition with bound hydrolysis products before accomplishing the power stroke.

## 1. Introduction

The ATP-driven mutual interplay between myosin and actin drives muscle contraction and many other forms of cellular motility, including cytokinesis and directional cargo transport; yet, our molecular understanding of how these two cytoskeletal proteins communicate with each to generate force still remains incomplete [1,2]. In a simplified view, actomyosin-based force production can be described with a four-state model, firstly formulated by Lymn and Taylor [3]. The model qualitatively illustrates the nucleotide-dependent conformational transitions of myosin during the cyclic traversing from actin-attached to actin-detached states (Figure 1a). While strongly bound to actin (rigor state), ATP binding to the myosin motor domain weakens the association with actin by a several 1000-fold decrease in affinity [4] and the motor dissociates rapidly and irreversibly adopting an intermediate state known as post-rigor [5,6,7]. Preceding ATP hydrolysis, the neck region of myosin—an elongated α-helical region adjacent to the motor domain with bound light-chains known as lever-arm—is set to an up position. The conformation is referred to as the pre-power-stroke state [8,9,10]. The myosin subsequently reattaches to actin with increasing affinity, adopting a structurally yet not assessed intermediate conformation assigned as the start-of-power-stroke state [11,12]. The existence of this state has been predicted from kinetic and modeling data and postulated to be required for strengthening the association with actin, which in turn triggers product release and forces myosin to perform the key event of force production, the power stroke—a swinging motion of the lever-arm from the up to the down position [11,13,14,15,16,17,18]. 

While there is still lack of high-resolution structures of conformational intermediates that could provide insights into how actin triggers product release and how distant sites allosterically communicate to allow myosin to achieve strong actin binding for initiating the power stroke, kinetic data have provided an almost quantitative description of the predominant equilibrium states of the actomyosin ATPase cycle [19] (Figure 1b). Key features of the cycle are the near irreversible nature thermodynamically manifested in a reciprocal relationship between myosin’s affinity for actin and nucleotides [4,20] and the activating role of actin in accelerating product release [21]. The kinetic investigations indicate that force is generated upon phosphate (P_i_) release concomitantly with the closing of the actin-binding cleft of myosin, which strengthens the actin interaction and initiates the power stroke [22]. The cleft-closed conformation on the other hand is essential for facilitating the release of the hydrolysis products. The weak-to-strong actin binding transition is supported by structural data from X-ray crystallographic and cryo-electron microscopy studies, which resolved cleft-open and cleft-closed conformations and uncovered molecular details of the interaction interface of the proteins [23,24,25,26,27]. Together with computational modeling, the structural data reveal that local conformational rearrangements of the active site elements switch-1, switch-2, and P-loop upon ATP binding (Figure 1c), are amplified to larger movements of structural elements and subdomains, forcing myosin to open the cleft [28,29,30], which reduces the contact area and weakens the affinity to actin; this is important for priming the lever-arm, a process referred to as the recovery step [31]. Cleft opening is accomplished by a rotational movement of both the L50 kDa domain (~16°) and the U50 kDa domain (~25°) causing the full dissociation of the actomyosin complex [18,32,33,34,35,36]. Recent work established the structural basis of the two-step mechanism of ADP release associated with a lever-arm swing that proceeds through a larger and subsequent smaller rotational movement [15,37,38]. 

The molecular events that enable myosin to bind to actin in a post-hydrolysis state with bound ADP and P_i_ are still a matter of debate [11,13,14,17,39]. The state is predicted to represent a high actin affinity conformation with the actin-binding cleft closed; however, it is different from the classical rigor state with the lever-arm in an up position [20,40,41]. 

All these studies concerning sub-processes of the actomyosin cycle imply a tight allosteric coupling between the nucleotide-binding site, the actin-binding region, and the converter domain; however, despite a detailed description of the underlying equilibrium states, the allosteric pathways in the myosin motor involved in force-production are only partially understood [42,43]. *In silico* modeling of the start-of-power-stroke structure and molecular dynamics simulation suggests, that cleft closure could be achieved without opening of switch-1, to prevent product release, but through a twisting of the central β-sheet that induces relay bending and converter rotation [11].

A critical role in controlling the sequence of events associated with force production appears to be played by the W-helix, which we propose to act as kind of communication zone in the inner myosin core for coupling conformational rearrangements upon nucleotide binding to cleft closure-induced β-sheet distortions by putting a stress on β1 to β3 of the central transducer; this is important to temporally couple weak-to-strong actin-binding transitions to the initiation of the power stroke and release of hydrolysis products. In-depth analysis of the model structure allowed us to identify hotspot residues T647 and T648 (*D. discoideum* myosin-2 numbering) in the connecting loop (W-loop) that appear to be critically involved in controlling structural changes during actin interaction and pass them on to execute the working stroke. These residues were identified in a series of targeted molecular dynamics simulations, which were used to mimic the effect of actin-induced cleft closure and the associated allosteric coupling mechanisms in the myosin motor domain [11]. Polar interactions of the two threonine residues stabilized the W-loop that connects the W-helix and β3 of the transducer during the weak-to-strong transition along the TMD simulations. The importance of the W-helix for force output is underpinned by the occurrence of myopathy mutations in this region associated with heart failure [44]. 

To gain mechanistic insights into the proposed role of the W-helix/transducer region in mediating actin binding to product release, we followed a mutational approach and replaced the two hotspot threonines in the W-loop of the myosin-2 motor domain from *D. discoideum* by either two alanines (AA) or three glycines (GGG) in order to reduce the rigidity of that region during weak-to-strong transition without dramatically disturbing the fold of the protein. The corresponding constructs are assigned as wild-type (M765^wt^) and mutants (M765^AA^, M765^GGG^), and minimized the stabilizing interactions of the W-helix with the β3-strand. The mutations were selected as rationalized from targeted molecular dynamics simulations that are predicted to affect the flexibility of the loop in such a way that the W-helix loses the ability of putting stress on the central transducer, thereby decoupling cleft-closure from β-sheet twisting. Replacing the two W-loop residues by three glycines in M765^GGG^ was expected to affect loop flexibility more drastically than with the double-alanine mutant construct M765^AA^. The structural disturbances introduced by these mutations are expected to influence the allosteric propagation of events from the hydrolysis-competent detached pre-power-stroke state to the end of the power stroke. By choosing an approach based on comprehensive transient kinetic analyses of the entire ATPase cycle between wildtype and mutant myosins, our data demonstrate that the W-helix couples actin-binding to product release. Moreover, the observed alterations in the duty ratio of the triple glycine mutant M765^GGG^ and its preferred population of actin-bound states, in which the rate of product release limits the actomyosin ATPase cycle, support the hypothesis, that the power stroke is initiated during a strongly bound actin state of myosin, in which the products are formed but not released.

## 2. Results and Discussion 

We crystallized the motor domains of the mutant myosins M765^AA^ and M765^GGG^ in complex with ADP∙VO_3_^2−^ and solved the atomic structures of the proteins to a resolution of 2.60 Å and 2.55 Å, respectively (Table 1). The superimposed structures reveal that the mutants crystallized in the pre-power-stroke state (Figure 2a) adopting the same overall conformation as previously reported for the wildtype (Smith and Rayment, 1996). Characteristic features of this pre-power-stroke state are the partially open actin-binding cleft, the up position of the converter, and the relay helix in a bent conformation. The nucleotide position and nucleotide-binding pocket are conserved between wild-type and mutants. The active site elements, switch-1 and switch-2, adopt the characteristic closed conformations and coordinate the nucleotide together with the P-loop (Figure 2b). Only minor structural deviations from the wild-type conformation can be observed for the mutants concerning the β3, β5, and β7 strands of the central transducer, which are slightly shifted, as well as the relay helix conformation (Figure 2c). 

The high resemblance of the wild-type and mutant structures, which are representative for the actin-detached conformation of myosin just prior to P_i_ release, and actin-rebinding suggest that the mutations have no or only a minor impact on ATP turnover if actin is absent, consistent with our hypothesis that the W-helix primarily mediates the communication between active site elements and the actin-binding region. Steady-state ATPase measurements with the purified proteins do lend initial support to the validity of our assumption. In the absence of actin, we observed normal, wild-type-like ATPase activity for both mutants (*k*_basal_), while in the presence of actin, pronounced alterations in actin-activated ATP turnover were obtained most pronounced for M765^GGG^ (Figure 3a, Table 2). The extended W-loop in the M765^GGG^ mutant led to an increase in the maximum ATP turnover at saturating actin concentrations (k_cat_) by almost threefold, yielding a rate of 7.8 ± 0.5 s^−1^ compared to 2.6 ± 0.5 s^−1^ and 4.1 ± 0.5 s^−1^ for wild-type and M765^AA^ mutant, and a more than twofold increase in the affinity for F-actin in the presence of ATP, as defined by K_app_ (i.e., the concentration of F-actin at k_cat_/2) of 54 ± 5 µM. Comparison of the catalytic efficiencies (*k*_cat_/*K*_app_) as a representative measure for the second-order rate constant for actin binding in the presence of ATP (Table 2), reveals that ternary complex formation (A∙M∙T) and product release in M765^GGG^ are considerably accelerated. 

To investigate the functional consequences of the altered ATPase of M765^GGG^, we performed *in vitro* motility assays (Figure 3b). Since all constructs were equipped with an artificial lever composed of two α-actinin repeats [45], we were able to directly compare motor performances using standard *in vitro* motility assays independent of potential influential effects induced by the light chains [46]. M765^GGG^ displayed a fourfold reduced actin translocation speed (*v*_avg_ = 0.13 ± 0.01 µm∙s^−1^) compared to 0.52 ± 0.02 µm∙s^−1^ and 0.54 ± 0.01 µm∙s^−1^ for M765^wt^ and M765^AA^, respectively. Two scenarios are conceivable for the reduced motility. Either the mutation affects a pathway that prevents a full swing of the lever-arm, e.g., by disturbing converter rotation, which would reduce the working stroke and consequently result in slower motility or the mutation influences individual kinetic steps in the cycle that affect the motile behavior, e.g., population of strongly actin-bound states [47]. 

To clarify the mechanism underlying the weak motor performance of the M765^GGG^ mutant, we conducted in a first set of experiments single ATP turnover measurements and estimated the effectiveness by which products were released. By mixing myosin with sub-equimolar concentrations of fluorescently labeled ATP (mATP) in the absence or presence of excess F-actin, we observed triphasic transients for all constructs (Figure 3c,d). The initial increase in fluorescence reports the kinetics of mATP binding to myosin with the corresponding rate *k*_on_. The following plateau phase covers the time range of bound hydrolysis products and the subsequent decay phase describes the kinetics of product release (*k*_off_). In the absence of actin, only minor difference in the overall course of the reaction between wild-type and mutants was observed. However, in the presence of actin, the rates of ATP binding (***k’*_on_**) and product release (***k’*_off_**), as obtained from three-exponential fitting to the transients displayed highly accelerated kinetics (Table 2). 

The effect of actin on both, nucleotide binding and product release was estimated from the ratio of ***k’*_on_**/*k*_on_ and ***k’*_off_**/*k*_off_ revealing more than fourfold and 10-fold accelerated kinetics for M765^GGG^, respectively. For comparison, M765^wt^ and M765^AA^ displayed an extended ADP release phase, rate-limited by P_i_ release during the weak-to-strong actin-binding transition [48,49]. The accelerated kinetics of the third phase (***k’***_off_ = 0.20 ± 0.01 s^−1^) suggest that the structural disturbances of the W-loop in M765^GGG^ affect the weak-to-strong transition towards an accelerated P_i_ release. 

To validate our conclusion that the mutations do not influence single steps of the actin-free ATPase of myosin, we determined rates and equilibrium constants underlying the interaction with nucleotides alone (Figure 4a–f; Table 3). Additionally, we studied actin interactions in the absence of nucleotides and in the presence of ADP (Figure 4g,h; Table 4). In agreement with the above steady-state data, the kinetics of ATP binding, hydrolysis, P_i_ release, ADP binding and dissociation were largely unaffected by the mutations in the absence of actin. The data reveal a minor involvement of the W-helix in the actin-free myosin ATP cycle. 

However, under physiological conditions, where myosin transverses through actin-dependent pathways of nucleotide interactions, we observed pronounced changes in the equilibria and kinetics for individual steps of the ATPase cycle. According to the scheme in Figure 5a, M765^GGG^ displayed accelerated ATP-induced actomyosin dissociation kinetics (Figure 5b,c) reflected by an almost fourfold increase in the second order rate constant of ATP binding to actomyosin (***k’*_1_*k’*_+2_** = 0.92 ± 0.01 µM^−1^s^−1^). Additionally, the data suggest that M765^GGG^ has a stronger preference for populating the A∙M∙T collision state as indicated by an almost fourfold increase in the corresponding equilibrium constant (1/***K’*_1_** = 883 ± 30 µM). The equilibrium from the ternary A∙M∙T state to low affinity A-M*∙T state is, in comparison to wild-type, favored towards the A∙M∙T state as indicated by a more than 20-fold increase in the first order rate constant ***k’*_−2_** underlying the isomerization equilibrium (Table 4). The strongly actin-bound states of myosin are of particular importance, since their duration and fractional occupancy are the main determinants of the duty ratio—the fraction of time the motor spends strongly bound to actin relative to the total duration of its ATPase cycle time. With a few exceptions, class-2 myosins are low-duty-ratio motors characterized by a fast and not rate-limiting ADP release step [46,50,51,52].

To enable the calculation of the duty ratio, we determined the equilibrium parameters of the strongly bound states by the ADP-inhibition of the ATP-induced actomyosin dissociation reaction (Figure 5d–g), from which the actomyosin affinity for ADP and the rate of ADP release can be deduced (Table 4). Both, wildtype and M765^AA^ displayed single exponential dissociation kinetics at all ADP concentrations used (Figure 5d). The corresponding affinities of ADP for A∙M and the rates of ADP release rates were similar (Figure 5e). For M765^GGG^, however, the ATP-induced A∙M dissociation in the presence of ADP followed a biphasic behavior (Figure 5f), which is indicative of a highly favorable A∙M∙D state, where ADP and ATP strongly compete for the same binding site [53]. The corresponding amplitudes of the fast and the slow phase displayed a reciprocal hyperbolic dependency on ADP concentration (Figure 5g). The non-linear regression revealed a more than 50-fold higher affinity (1/(***K’*_5_*K’*_6_**) = 2.2 ± 1 µM) for M765^GGG^ compared to wildtype and M765^AA^ (Table 4). 

This unexpected high ADP-affinity of M765^GGG^ in the actin-bound state prompted us to repeat the ADP-inhibition experiments using reduced levels of ATP, which allowed us to determine the second order rate constant of ADP binding to actomyosin (***k’***_−5_***K’***_6_) and the rate of the transition from A∙M∙D to A∙M-D (***k’***_+5_), which determines the rate of ADP release [54]. The biphasic transients (Figure 5h) followed a hyperbolic dependence with respect to rate of the slow phase (*k*_obs_^slow^) (Figure 5i) and a linear dependence with respect to rate of the fast phase (*k*_obs_^fast^) (Figure 5i, inset). Linear and non-linear fitting yielded the parameters ***k’*_+5_** and ***k’*_−5_*K’*_6_** summarized in Table 4, revealing more than a sixfold deceleration in ADP release kinetics for M765^GGG^ (***k’*_+5_** = 16 s^−1^), in a time range typically observed for high-duty-ratio class-5 myosins [55,56,57].

To assess the rate-limiting step in the cycle, we first determined the rate of P_i_ release from actin-bound myosin using sequential mixing experiments (Figure 6a) and MDCC-PBP as P_i_ sensor [21]. Under multiple turnover conditions (excess ATP), M765^GGG^ displayed accelerated P_i_ release kinetics in comparison to wildtype (Figure 6b). The observed rate constants followed a hyperbolic dependency with increasing actin concentrations according to *k*_obs_ = ***k’*_+4_**[actin]/(1/***K’*_DPA_**+[actin]) (Figure 6c). At saturating actin concentrations, the rate of P_i_ release was ***k’*_+4_** = 14.4 ± 1.0 s^−1^ and the affinity of the M∙D∙P_i_ state for actin, as deduced from the hyperbolic fit, was fivefold higher (1/***K’*_DPA_** = 18 ± 3 µM) than the corresponding values for wild-type ***k’*_+4_** = 3.8 ± 1.1 s^−1^ and 1/***K’*_DPA_** = 94 ± 32 µM. The second order rate constant of M∙D∙P_i_ binding to actin defined by ***k’*_+4_*K’*_DPA_** was increased by more than 20-fold for M765^GGG^ (Table 4). 

The same actin titration experiments performed under single turnover conditions showed a biphasic behavior for both, wild-type and M765^GGG^ (Figure 6d). The fast phase was dependent on actin, which allowed the determination of the P_i_ release rate (***k’*_+4_**) under saturating actin conditions. This was obtained from the hyperbolic dependency of the observed rates according to *k*_obs_ = ***k’*_+4_**[actin]/(1/***K’*_DPA_**+[actin]) (Figure 6e). Under these single turnover conditions, the kinetics of the slow phase are limited by the rate of ATP binding (since low ATP concentration were used) and are thus independent of actin (Figure 6g). Both, single and multiple turnover experiments reveal a comparable, actin concentration-dependent steep increase in the actin-activated P_i_ release kinetics. The initial slope provides a measure for the second order binding constant of actin to M∙D∙P_i_. The data reveal almost 20-fold increased second order rate constants of M∙D∙P_i_ binding to actin (***k’*_+4_*K’*_DPA_**) for M765^GGG^ (Figure 6c,e). Plots of the corresponding amplitudes of the fast and slow phases under single turnover conditions reveal a reciprocal hyperbolic dependency on actin for wildtype and M765^GGG^. The amplitudes report the amount of liberated P_i_.

Estimates of the fractional occupancy of the actin-attached states of myosin and corresponding kinetics underlying weak-to-strong transitions were assayed by two different experimental approaches. First, we performed sequential mixing experiments, where 2 µM myosin was rapidly mixed with sub-equimolar concentrations of ATP (1.8 µM), aged for 3s to allow ATP binding and hydrolysis, and then mixed with 20 µM pyrene-labeled actin (Figure 6h). The observed fluorescence quench reports the transition from the high-fluorescence weak-binding state to the low-fluorescence strong-binding state [58] (Figure 1b). For wild-type, the observed rate of weak-to-strong transition was 1.0 ± 0.1 s^−1^ and for M765^GGG^ it was 2.3 ± 0.2 s^−1^. The rates are in the range of the steady-state ATPase at the defined actin concentration (10 µM), revealing that the weak-to-strong transition limits the ATPase rate in both myosins; however, for M765^GGG^ the transitions proceeds more than twofold faster. 

We then recorded the entire process of actomyosin dissociation and re-association by mixing a preformed actomyosin complex with excess ATP (Figure 6i). The obtained transients were normalized to the maximum amplitude of fluorescence quench, which corresponds to the amount of actin-bound myosin in the absence of nucleotides. The initial decrease in fluorescence reflects the kinetics of the actomyosin dissociation and the subsequent increase in fluorescence describes the post-hydrolysis process of re-association. The dissociation rates between wild-type and mutant differ by threefold as well as the association rates, which indicates that M765^GGG^ dissociates faster, but with smaller amplitude and also re-associates faster (Table 4). Since the dissociation kinetics describes the conformational transition of myosin from the actin-bound state to the detached state and the association kinetics the rebinding through weak-to-strong transitions, the fractional occupancy of the strongly bound states can be extracted from the change in total amplitude (Table 4). This fraction provides also an estimate for the duty ratio at the defined actin concentrations [56]. In agreement with the calculated duty ratio from the kinetic data in Table 4, the strongly actin-attached states of M765^GGG^ cover ~45% of the cycle, compared to less than 1% in the case of the wildtype.

To corroborate the interpretation of the results, we simulated the fractional occupancy of the main intermediate ATPase states of the cycle (Figure 7a) using the experimentally obtained kinetic parameters and equilibrium constants exemplary for four actin concentrations ([A] = 0.1*K*_app_, 1*K*_app_, 3*K*_app_ and 20*K*_app_) under conditions of excess ATP (10 mM). The simulations have previously been applied to allow comparisons of the state occupancies between fast and slow muscle myosin isoforms [59]. The parameters used for the simulations are summarized in Table 5. The results are shown as pie charts in Figure 6b with the corresponding values listed in Table 6. 

As expected, state occupancies were actin-dependent, visible in the obvious fractional shift from detached (M∙T and M∙D∙P_i_) to attached states as the actin concentration increased. The strongly attached states and the duty ratio increased accordingly. However, M765^GGG^ displayed a state distribution in favor of the population of the strongly bound A∙M∙D state and disfavor of the weakly attached A-M∙D∙P_i_ state. The fractional occupation of the A∙M∙D state increased from 3.3% at [A] = 0.1 × *K*_app_ to 35.9 % at [A] = 20 × *K*_app_. For wild-type structures, the A∙M∙D state occupancy was less pronounced and increased from 0.3% to 3%. This alteration in the weak-to-strong transition is the result of a combination between accelerated actin-activated P_i_ release and higher ADP affinity of the A∙M∙D-state as consequence of a considerable reduction in the rate of ADP release. Together, the results are in agreement with a higher cycling speed, increased duty ratio, and slower actin translocation speed. For all constructs, the experimentally determined actin translocation velocities fully agree with the calculated values from the simulation (Table 6).

## 3. Conclusions

The functional characterization of the mutants clearly supports the hypothesis of a W-helix-mediated communication pathway in the myosin motor that affects product release. The replacement of the threonines by glycines in M765^GGG^ led to an enhancement of the actin-activated steady-state ATPase, while motor performance was drastically reduced. The slow motility results primarily from the increased duty ratio as a consequence of rate-limiting P_i_ and ADP release; the latter prolongs the strongly actin-bound myosin states relative to a single ATP turnover. The kinetics underlying the accelerated P_i_ release and decelerated ADP release in M765^GGG^, which are in terms of the corresponding rate constants almost identical (***k’*_+4_** = 14.4 s^−1^ and ***k’*_+5_** = 16 s^−1^) with a total life time of τ_release_ = τ***’*_+4_** + τ***’*_+5_** ≈ 132 ms that limits the entire ATPase cycle time (τ_total_ ≈ 128 ms), can be explained by a mechanism, in which actin triggers structural transitions associated with P_i_ release more effectively than the wild-type, apparently through a changed actin interface that strengthens the actin interaction, likely a closed-cleft conformation of high actin affinity. Such a state resembles the previously postulated post-hydrolysis start-of-power-stroke conformation of myosin shown in Figure 7a (A∙M∙D∙P_i_), where the cleft is closed, the hydrolysis products are bound, and the lever-arm is still in the up position [11]. The data are consistent with the model that cleft-closure precedes P_i_ release [60]. On the other hand, actin-induced structural transitions associated with ADP release are disrupted by the mutations. This is reflected in the high ADP affinity of actomyosin (1/***K’*_5_*K’*_6_**) and the reduced ADP release rates from actomyosin (***k*’_+5_**) (Table 4). 

This interpretation of our data correlates well with the prediction that the mutation-induced W-loop disturbance decouples cleft-closure from ADP release, consistent with the view that the power stroke is initiated during the transition from the initial weak binding state to the strongly actin-bound state followed by a highly irreversible P_i_ release step and a slower second step associated with ADP release [61]. This pathway of events is also supported by the kinetic signature of M765^GGG^. The mutant displays rates of P_i_ and ADP release that depend reciprocally on actin, where P_i_ release is accelerated and ADP release is drastically decelerated. The unusual kinetics require that actin triggers product release serial through subsequent steps of communication. In a first step, actin binding enables the opening of an escape route for P_i_, which we speculate to occur during the weak-to-strong transition and by the adaptation of a cleft-closure conformation (A∙M∙D∙P_i_) with the lever-arm up (Figure 7a). We assume that this start-of-power-stroke state accommodates all necessary structural features to enable a fast and highly irreversible release of P_i_ for effectively driving the reaction to the next state associated with ADP release in a second, smaller lever-arm swing [15]. Actin-activated ADP release, however, is drastically reduced in M765^GGG^. The structural disturbances in the W-loop appear to affect the W-helix/transducer region, which acts as a central communication hub between nucleotide and actin-binding sites, but they also could cause distortions in loop-2, which is connected to the W-helix and one of the major actin-binding loops of the upper cleft implicated in regulating ADP release by modulating the strength and duration of the actin-attached states [62,63]. The role of the W-helix in this context is to coordinate the allosteric propagation of events from the hydrolysis-competent detached pre-power-stroke state to a strongly attached state, thereby determining the extent of the fractional occupancies of the strongly actin-attached states. From the structural point of view, the W-helix appears to serve dual functions: coupling nucleotide state to cleft-closure and mediating the twisting of the transducer to the relay helix, thus linking actin binding to product release and converter rotation. Given the structurally poorly defined states prior and subsequent to the power stroke, the mutant is a promising candidate for future investigations to obtain structural details of the actin-attached states of myosin associated with the power stroke.

## 4. Materials and Methods 

### 4.1. Plasmid Construction and Protein Purification 

Expression plasmids pDXA-M765^AA^-2R and pDXA-M765^GGG^-2R, encoding motor domain constructs of *D. discoideum* myosin-2 (amino acids 1-765) with mutations T647A, T648A and A647G, A648G, A649G, respectively, as fusion to an artificial lever consisting of two alpha-actinin repeats (2R) and a C-terminal His_8_-tag, were obtained by two consecutive PCR reactions using a combination of primers 5′-CAACCCTCGAGGCAGCCAACCCACAT-3′ and 5′-CAACCCTCGAGGGAGGCGGCAACCCACAT-3′ with primer 5′-GGTAAAACTTGAATTGATCCTCTAG-3′ and the pDXA-M765 plasmid as template. The PCR products were then applied to a second PCR reaction with primer 5′-GTACCGAGGATCCAATTCATG-3′ as forward primer to amplify the motor domains. PCR products were subsequently cloned into the pDXA-2R vector using BamHI and NsiI restriction sites to obtain the expression plasmids encoding M765^AA^-2R and M765^GGG^-2R. Constructs were verified by sequencing. Wild-type and mutant constructs were produced in *D. discoideum* and purified as described [48]. Chicken skeletal actin was purified as described [64]. Pyrene-labeled actin was prepared from skeletal actin as described [58]. Phosphate binding protein from *E. coli* was recombinantly purified from *E. coli* Rosetta (DE3) pLysS cells and labeled with 7-Diethylamino-3-[N-(2-maleimidoethyl)carbamoyl]coumarin (MDCC) as described [65].

### 4.2. X-ray Crystallographic Analysis 

Crystals of the mutant M765^AA^ and M765^GGG^ myosins were obtained by co-crystallizing the constructs with 2 mM ADP, 2 mM metavanadate and 2 mM MgCl_2_ at 4 °C using the hanging drop vapor diffusion method. The mixture was pre-incubated for 30 min and subsequently mixed with an equal volume of reservoir containing 0.24 M sodium malonate pH 6.5–8.0 and 21%–27% PEG 3350. Crystals were cryoprotected with ethylene glycol prior to data collection at synchrotron beamlines P13 at DESY (Hamburg, Germany) and Proxima-2A at SOLEIL (St. Aubin, France). The datasets were processed with XDS [66] and scaled with AIMLESS [67] from the ccp4 software suite [68]. Molecular replacement using the *D. discoideum* myosin-2 pre-power-stroke structure (pdb: 1vom) [8] as starting model was carried out with phaser [69]. Final model building and structure refinement was performed using Coot [70] and phenix.refine [71]. The final model and structure factor amplitudes were deposited in the RCSB PDB Protein Data Bank (www.rcsb.org) [72] with accession codes M765^AA^ (pdb:7B1A) and M765^GGG^ (pdb:7B19). Refinement statistics are listed in Table 1.

### 4.3. Kinetic Experiments, Simulations, and In Vitro Motility Assays 

Buffers and chemicals were obtained from Thermofisher Scientific Inc., Waltham, MA, USA. Steady-state ATPase measurements were performed at 25 °C in buffer containing 25 mM 2-(4-(2-Hydroxyethyl)-1-piperazinyl)-ethansulfonsäure (HEPES-KOH), 25 mM KCl, 5 mM MgCl_2_, 1 mM DTT and 2 mM ATP at pH = 7.3 using the NADH-coupled assay [63]. Unless otherwise stated, transient kinetic measurements were performed at 20 °C in experimental buffer containing 20 mM 3-(N-morpholino)propanesulfonic acid (MOPS), 100 mM KCl, 5 mM MgCl_2_ and 1 mM DTT at pH = 7.0 using the Hi-tech Scientific SF-61DX double-mixing stopped-flow system (TgK Scientific Limited, Bradford on Avon, UK). Populations of the intermediate ATPase states were calculated with the MUSICO software [59]. Simulations were performed with different concentrations of actin, 5 mM ATP and the concentrations of P_i_ and ADP were set to zero. Sliding velocities of tetramethylrhodamine isothiocyanate (TRITC) phalloidin-labeled actin filaments were measured in experimental buffer (25 mM imidazole, 25 mM KCl, 4 mM MgCl_2_, 1 mM EGTA, 10 mM DTT, 0.05% BSA, 2 mM ATP, 30 mM glucose, 10 U/mL glucose oxidase, 1 kU/mL catalase and 0.2% methylcellulose, pH = 7.4) at 23 °C as described [55]. Penta∙His Antibody (Qiagen GmbH, Germany) was used for the specific attachment of the wild-type and mutant myosin motor domain fragments on nitrocellulose-coated coverslips. Actin filament tracking was performed using DiaTrack 3.05 software. Sliding velocities were obtained from the Gaussian distributions of three independent experiments and analyzed using Origin 2018b software (OriginLab, Northampton, MA, USA). 

## Figures and Tables

**Figure 1 ijms-22-00104-f001:**
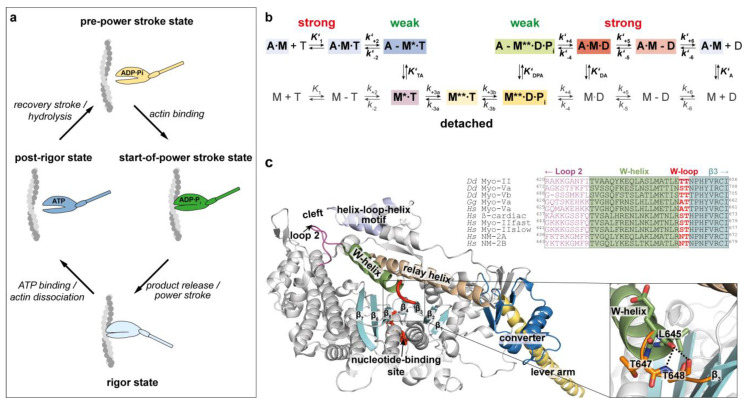
The actomyosin ATPase cycle and structural features of the myosin motor domain. (**a**) The Lymn–Taylor four-state model and the predominant equilibrium states of the actomyosin cycle. Rigor state: strongly actin-bound state (lever arm in “down” position). Post-rigor state: actin-detached state with bound ATP (actin-binding cleft “open”). Pre-power-stroke state—weakly actin-bound state; conformational state after the recovery step with hydrolyzed ATP; lever arm in “up” position. Start-of-power-stroke state: myosin re-associates with actin (binding triggers product release and initiates the power stroke). (**b**) Kinetic scheme of the actomyosin ATPase reaction. Colored states illustrate nucleotide-associated conformational transitions of myosin. A: actin; M: myosin; M*, M**: fluorescence sensitive states of myosin; T: ATP; D: ADP; P_i_: phosphate. Strong nucleotide and actomyosin interactions are indicated with a dot, weak interactions with a hyphen. Equilibrium constants and rate constants depending on actin are given with a prime in bold italic (e.g., ***K’_1_*, *k’_+1_***), all others in italic font style (e.g., ***K’*_1_**, ***k’*_+1_**). Equilibrium constants indexed with T, D, P, and A represent the affinity of the respective nucleotide state of myosin for actin (e.g., ***K’*_DPA_**). (**c**) Structural elements and domains in the myosin-2 motor involved in coupling actin binding to force generation. X-ray crystal structure of the *D. discoideum* myosin-2 motor domain (PDB: 1g8x) highlighting the W-loop in red as connecting element of the W-helix (green) and the transducer (cyan) and close-up view of the W-helix/transducer region showing amino acids T^647^ and T^648^ in the W-loop interacting with surrounding residues (inset). The W-helix interacts with β-sheets strands 3 to 7 (cyan), the relay helix (orange), the helix-loop-helix motif (blue) and the two connector sequences, the strut (yellow) and loop 2 (magenta). Multiple sequence alignment of the W-helix/transducer region for different myosin isoforms highlights the conservation of the W-loop throughout myosin classes.

**Figure 2 ijms-22-00104-f002:**
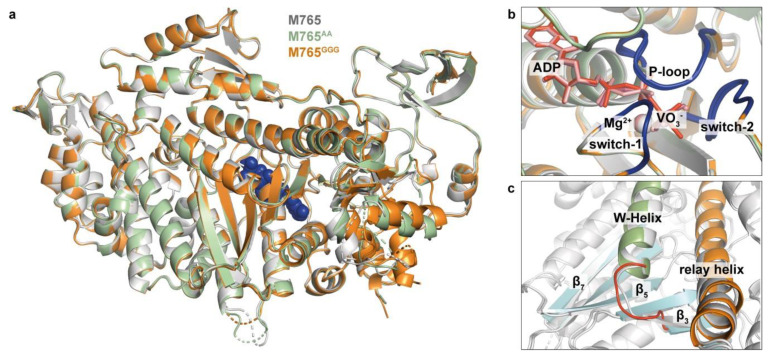
X-ray crystal structures of the *Dd* myosin-2 motor domains of wildtype myosin M765 (pdb:1VOM) and mutant myosins M765^AA^ (pdb:7B1A) and M765^GGG^ (pdb:7B19) in complex with ADP∙VO_4_^2−^. (**a**) Superposition of the structures. M765 (white), M765^AA^ (green), M765^GGG^ (orange). The nucleotide is shown in blue. (**b**) Close-up view of the nucleotide-binding pocket. The active site elements switch-1, switch-2, and P-loop adopt the same conformation in wildtype and mutants. (**c**) Close-up view on the W-helix/transducer region. In the M765^GGG^ structure (orange) small conformational shifts are visible for the β3, β5, and β7 sheets of the transducer (cyan) and the C-terminal end of relay helix (orange). The longer W-loop in M765^GGG^ prevents proper β3-sheet formation.

**Figure 3 ijms-22-00104-f003:**
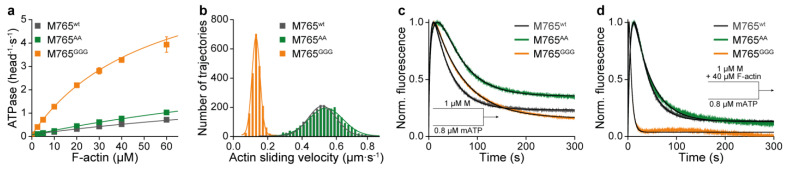
Motor properties, steady-state, and single ATP turnover kinetics. (**a**) ATPase activity of wild-type (M765^wt^) and mutant myosins (M765^AA^, M765^GGG^) as a function of actin. Rate constants of maximum ATP turnover (*k*_cat_) and affinity of myosin for F-actin in the presence of ATP (*K*_app_, i.e., the concentration of F-actin at which the ATPase rate is one-half of *k*_cat_). Parameters were obtained by fitting the data to hyperbolic functions according to Michaelis–Menten kinetics. (**b**) Actin filament translocation in *in vitro* motility assays. The average sliding velocity (actin translocation speed) for each myosin construct was obtained from Gaussian fits. (**c**,**d**) Single ATP turnover kinetics in the absence and presence of actin. Normalized fluorescence time traces upon mixing 1 µM myosin with 0.8 µM mATP in the absence (**c**) and presence of 40 µM F-actin (**d**).

**Figure 4 ijms-22-00104-f004:**
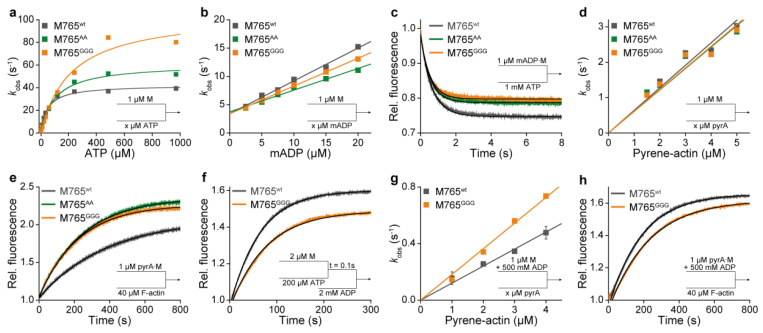
Interactions of myosin with nucleotides or actin. (**a**) Apparent second order rate constants of ATP binding to myosin (*K*_1_*k*_+2_) and ATP hydrolysis rate (*k*_+3_+*k*_−3_) were obtained from the hyperbolic dependence of *k*_obs_ on ATP. (**b**) Apparent second order rate constant of mADP binding to myosin (*k*_−5_*K*_6_) was obtained from the slope of the linear dependence of *k*_obs_. (**c**) Rate of ADP release (*k*_+5_) upon mixing 1 µM M∙mADP complex with excess ATP and fitting the transient to single exponential function. (**d**) Apparent second order rate constant of actin binding to myosin (***k’*_+A_**) was obtained from linear fits to the *k*_obs_ data. (**e**) Actin dissociation kinetics. An amount of 1 µM pyrene–actomyosin complex was mixed with excess unlabeled F-actin and the rate of actin dissociation (***k’*_−A_**) was obtained from exponential fits to the transients. (**f**) Rate of P_i_ release (*k*_+4_). An amount of 2 µM myosin was mixed with 200 µM ATP, aged for 0.1 s, and subsequently mixed with 2 mM ADP. All solutions contained 5 µM MDCC-PBP. (*k*_+4_) was obtained from exponential fits to the transients. (**g**) Apparent second order constant of actin binding to ADP–myosin (***k’*_+DA_**) was obtained from the linear dependencies of *k*_obs_ on pyrene-actin concentration. (**h**) Actin dissociation from the A∙M∙D complex. An amount of 1 µM pyrA∙M pre-incubated with excess ADP was mixed with excess unlabeled F-actin and the rate of actin dissociation (***k*_−DA_**) was obtained from single exponentials fits.

**Figure 5 ijms-22-00104-f005:**
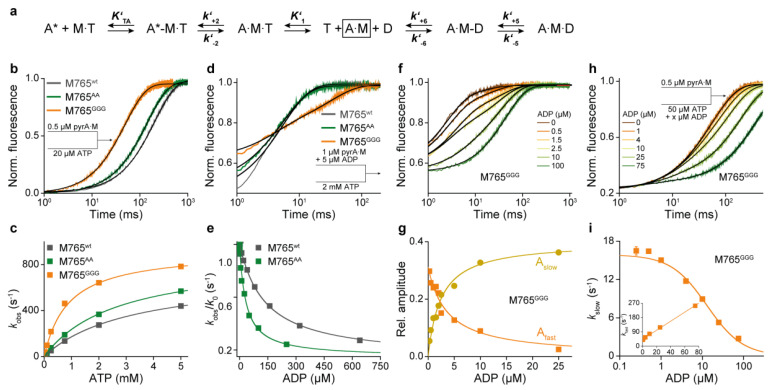
Myosin interactions with nucleotides in the presence of actin. (**a**) Kinetic scheme of the ATP-induced actomyosin dissociation. A*: pyrene-actin fluorescence; A: quenched pyrene-actin fluorescence with bound myosin. (**b**) Normalized fluorescence transients of the dissociation reaction upon mixing a complex of 0.5 µM pyrene-actomyosin (pyrA∙M) with 20 µM ATP. Single exponential fits to the transients yielded concentration-dependent rates (*k*_obs_) of ATP binding to actomyosin. (**c**) Second order rate constant of ATP binding to A∙M (***K’*_1_*k’*_+2_**) and rate of A∙M∙T to A-M∙T isomerization (***k’*_+2_**) were obtained from hyperbolic fits of the *k*_obs_ dependency on ATP. (**d**,**e**) Affinity of ADP to actomyosin (1/***K*’_5_*K*’_6_**) determined by mixing 1 µM pyrA∙M complex in the presence of increasing ADP concentrations with excess ATP. M765^wt^ and M765^AA^ displayed monophasic, and M765^GGG^ biphasic transients shown as exemplary for 2.5 µM ADP (**d**). (**e**) Plots of normalized rate constants (*k*_obs_/*k*_0_; with *k*_0_ = dissociation rate constant in the absence of ADP) against ADP concentration. Hyperbolic fits to the data yielded equilibrium dissociation constants of ADP from actomyosin (1/***K*’_5_*K*’_6_**). (**f**) For M765^GGG^, ATP-induced complex dissociation in the presence of increasing amounts of ADP could be fitted by double exponentials. (**g**) Reciprocal hyperbolic dependency of the relative amplitudes for the fast phase (A_fast_) and the slow phase (A_slow_) reveal the affinity of ADP for A∙M765^GGG^. (**h**,**i**) Competitive binding of ATP and ADP to A∙M765^GGG^. (**h**) Fluorescence transients upon mixing 0.5 µM pyrA∙M with a mixture of 50 µM ATP and increasing concentrations of ADP (0 to 75 µM). Transients were best fit by double exponentials, yielding rate constants for ADP release (***k’*_+5_**) from the y-intercept of the hyperbolic fit to the ADP-dependent rates of the slow phase (**i**). Liner fit to the ADP-dependent rate of the fast phase (**i, inset**) yields the second order rate constant of ADP binding to A∙M (***k’*_−5_*K’*_6_**).

**Figure 6 ijms-22-00104-f006:**
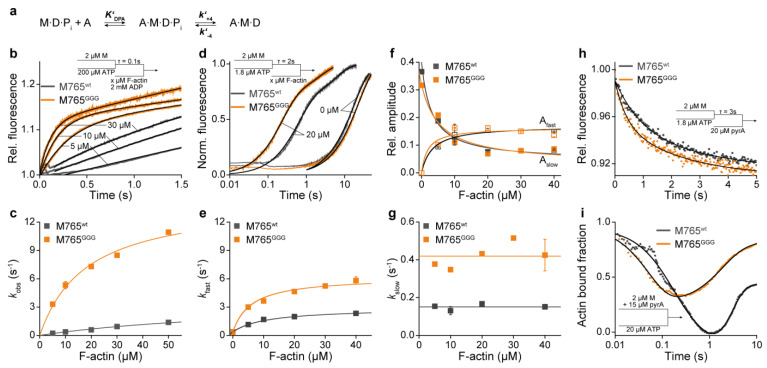
(**a**) Scheme of actin-activated P_i_ release during the weak-to-strong transition. (**b**) Rate of P_i_ release under multiple turnover conditions. An amount of 2 µM myosin was mixed with 200 µM ATP, aged for 0.1 s, and subsequently mixed with increasing F-actin concentrations in the presence of excess ADP. All solutions were supplemented with 5 µM MDCC-PBP as fluorescence-sensitive P_i_ sensor prior to mixing. The corresponding rates of the actin-activated P_i_ release were obtained from single exponentials fits including a slope term to account for the subsequent steady-state reaction. (**c**) Second order rate constants of actin binding to the M∙D∙P_i_ complex (***k*_+4_*K’*_DPA_**) and rate of P_i_ release (***k’*_+4_**). (**d**) Kinetics of P_i_ release monitored with 5 µM MDCC-PBP under single turnover conditions. The fast phase corresponds to the rate of P_i_ release and the slow phase to the rate of ATP binding, which limits the overall reaction rate. (**e**) Plots of *k*_fast_ vs. actin and hyperbolic fits to the data reveal second order rate constants of actin binding to the M∙D∙P_i_ complex (***k*_+4_*K’*_DPA_**). (**f**) Graph showing the reciprocal hyperbolic dependency of the relative amplitudes of the fast phase (A_fast_) and the slow phase (A_slow_) on actin associated with the amount of liberated P_i_ during a single ATP turnover. (**g**) Plot of the rate of the slow phase (*k*_slow_) vs. actin. (**h**) Kinetics of the weak-to-strong transition under single turnover conditions. An amount of 2 µM myosin was mixed with sub-equimolar ATP, aged for 3 s, and rapidly mixed with 20 µM pyrene-actin. Double exponential fits reveal two rate constants of actin association. (**i**) Actomyosin dissociation and association kinetics under multiple turnover conditions upon mixing a solution of 2 µM myosin and 15 µM pyrene-actin with 20 µM ATP. Transients report the fast dissociation of the complex, followed by the re-association reaction. Signals were normalized relative to the maximum fluorescence intensity of 1 that corresponds to 100% of myosin bound to actin.

**Figure 7 ijms-22-00104-f007:**
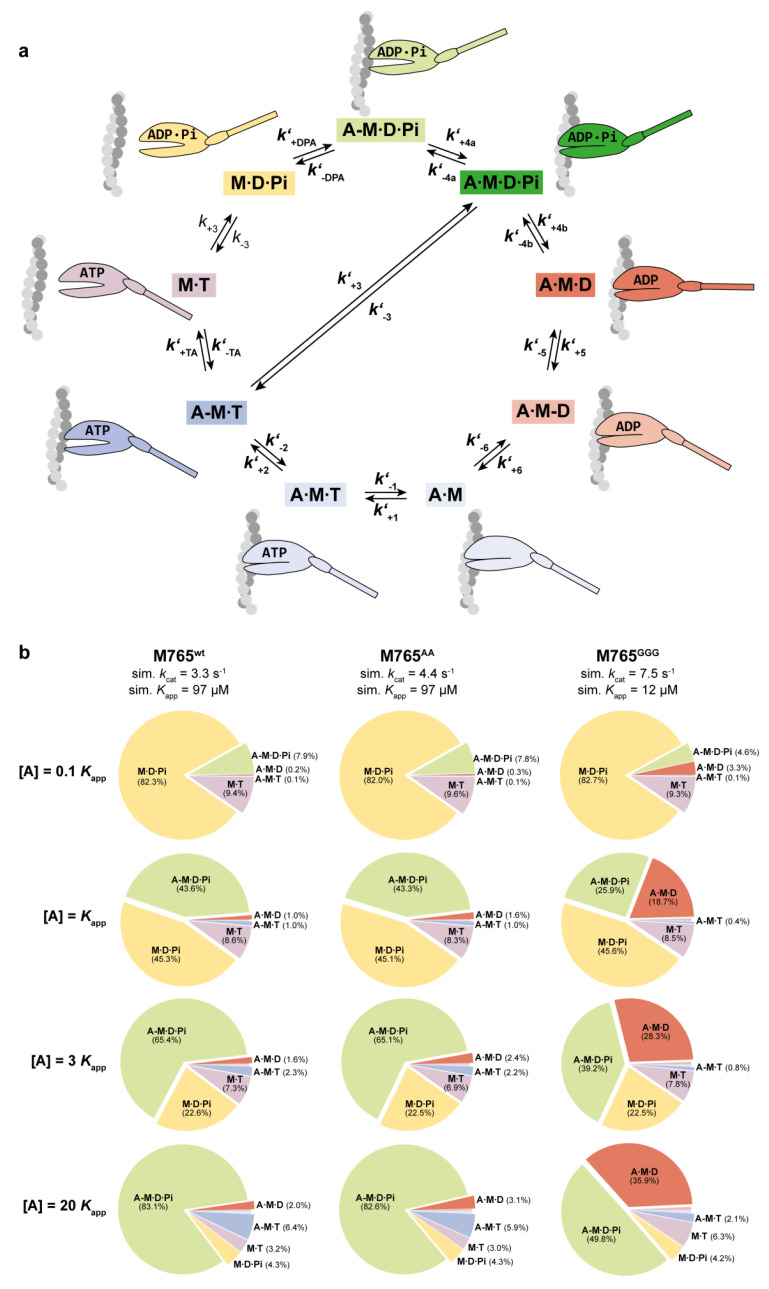
The actomyosin cross-bridge cycle and its major intermediate states. (**a**) A minimal, (9+1)–step model of the actomyosin ATPase cycle, which includes the start-of-power-stroke state (A∙M∙D∙P_i_ highlighted in green). Notations are the same as in Figure 2a. Structural states of myosin with weak actin affinity are depicted as globular heads with an open actin-binding cleft. Strongly actin-attached states are depicted with a closed actin-binding cleft, and detached states without contacting actin. Strongly actin-bound states (A∙M, A∙M∙T, A∙M∙D∙P_i_, A∙M∙D, A∙M-D), weakly actin-bound states (A-M∙T, A-M∙D∙P_i_) and detached states (M∙T, M∙D∙P_i_). (**b**) Pie chart illustration of the fractional occupancies of the ATPase cycle intermediates at different actin concentrations (0.1K_app_, 1K_app_, 3K_app_ and 20K_app_) for wildtype and mutants as obtained by modeling using the MUSICO program [59]. Modeling is based on experimentally determined parameters and complemented by estimated values as summarized in Table 5. The relative occupancy of the accessible intermediate states is shown as a percentile in the pie charts. The percentiles of the short-lived intermediates A∙M∙T, A∙M, and A∙M-D range for all constructs between 0.04% and 0.2%.

**Table 1 ijms-22-00104-t001:** X-ray data collection and refinement statistics.

	M765^AA^	M765^GGG^
**Data collection**		
Space group	C2221	C2221
Cell dimensions		
a, b, c (Å)	89.76, 146.47, 154.85	89.19, 149.03, 153.93
α, β, γ (°)	90.00, 90.00, 90.00	90.00, 90.00, 90.00
Resolution (Å)	43.11–2.60 (2.69–2.60)	43.40–2.55 (2.64–2.55)
R_merge_ [%]	0.037 (0.355)	0.055 (0.361)
I / σI	13.68 (1.96)	9.77 (2.09)
CC_1/2_	0.999 (0.747)	0.997 (0.844)
Completeness (%)	99.77 (99.84)	99.55 (99.85)
Redundancy	2.0	2.0
**Refinement**		
Resolution (Å)	43.11–2.60	43.40–2.55
No. reflections	31661 (3109)	33626 (3315)
R_work_/R_free_	20.94/26.26	20.64/25.21
No. atoms	5958	6129
Protein	5788	5906
Ligand/ion	74	129
Water	96	94
B-factors		
Protein	63.27	56.10
Ligand/ion	68.30	66.39
Water	45.87	41.83
R.m.s. deviations		
Bond lengths (Å)	0.007	0.006
Bond angles (°)	0.79	0.87

**Table 2 ijms-22-00104-t002:** Steady-state ATPase, actin translocation velocity, and single nucleotide turnover kinetics.

	Unit	M765^wt^	M765^AA^	M765^GGG^
**ATPase activity** ^1^				
*k* _basal_	(s^−1^)	0.09 ± 0.02	0.07 ± 0.01	0.08 ± 0.01
*k* _cat_	(s^−1^)	2.6 ± 0.5	4.1 ± 0.5	7.8 ± 0.5
K_app_	(µM)	>100	>100	54 ± 5
*k*_cat_/K_app_	(µM^−1^s^−1^)	<0.03	<0.03	0.14 ± 0.01
***In vitro* actin filament sliding velocities** ^1^				
*v* _avg_	(µms^−1^)	0.52 ± 0.02	0.54 ± 0.01	0.13 ± 0.01
**Single ATP turnover kinetics and coupling ratios** ^1^				
*k* _on_	(s^−1^)	0.24 ± 0.01	0.070 ± 0.01	0.27 ± 0.01
*k* _off_	(s^−1^)	0.03 ± 0.01	0.02 ± 0.01	0.02 ± 0.01
***k*’_on_ ^2^**	(s^−1^)	0.16 ± 0.01	0.23 ± 0.01	0.83 ± 0.02
***k*’_off_** ^2^	(s^−1^)	0.04 ± 0.01	0.03 ± 0.01	0.20 ± 0.01
***k’*_on_**/*k*_on_	-	0.7 ± 0.1	3.3 ± 0.1	3.1 ± 0.1
***k’*_off_**/*k*_off_	-	1.3 ± 0.1	1.5 ± 0.1	10 ± 0.1

^1^ at 20 °C; ^2^ with 20 µM F-actin.

**Table 3 ijms-22-00104-t003:** Transient kinetic parameters and equilibrium constants in the absence of actin.

	Unit	M765^wt^	M765^AA^	M765^GGG^
**ATP binding to myosin ^1^**				
*K* _1_ *k* _+2_	(µM^−1^s^−1^)	0.48 ± 0.01	0.38 ± 0.01	0.38 ± 0.01
*k* _−2_	(s^−1^)	0.79 ± 0.49	0.54 ± 0.24	0.43 ± 0.25
*k*_+3_+*k*_−3_	(s^−1^)	42 ± 1	61 ± 2	106 ± 8
**ADP binding to myosin ^1^**				
*k* _−5_ *K* _6_	(µM^−1^s^−1^)	0.58 ± 0.02	0.38 ± 0.02	0.49 ± 0.01
*k* _+5_	(s^−1^)	1.6 ± 0.1	1.8 ± 0.1	1.9 ± 0.1
*1*/(*K*_5_*K*_6_)	(µM)	2.8 ± 0.2	4.7 ± 0.4	3.9 ± 0.2
**Rate of P_i_ release ^1^**				
*k* _+4_	(s^−1^)	0.02 ± 0.01	n.d.	0.02 ± 0.01

^1^ at 20 °C.

**Table 4 ijms-22-00104-t004:** Transient kinetic parameters and equilibrium constants in the presence of actin.

	Unit	M765^wt^	M765^AA^	M765^GGG^
**Actomyosin interactions in the absence and presence of ADP**				
** *k’* _+A_ **	(µM^−1^s^−1^)	0.64 ± 0.03	0.61 ± 0.03	0.61 ± 0.03
** *k’* _−A_ **	(s^−1^)	0.003	0.004	0.005
***1***/***K’*_A_ ^1^**	(µM)	0.0057 ± 0.0005	0.010 ± 0.009	0.011 ± 0.009
** *k’* _+DA_ **	(µM^−1^s^−1^)	0.12 ± 0.01	n.d.	0.18 ± 0.01
** *k’* _−DA_ **	(s^−1^)	0.006 ± 0.001	n.d.	0.005 ± 0.001
***1***/***K’*_DA_ ^2^**	(µM)	0.063 ± 0.002	n.d.	0.024 ± 0.001
**ATP interactions of actomyosin**				
** *K’* _1_ *k’* _+2_ **	(µM^−1^s^−1^)	0.24 ± 0.01	0.32 ± 0.01	0.92 ± 0.01
** *k’* _+2_ **	(s^−1^)	719 ± 22	871 ± 80	923 ± 19
** *k’* _−2_ **	(s^−1^)	0.08 ± 0.05	0.62 ± 0.05	1.48 ± 0.13
** *1/K’* _1_ **	(µM)	3200 ± 130	2702 ± 279	883 ± 30
**ADP interactions of actomyosin**				
***k’*_−5_*K’*_6_ ^3^ **	(µM^−1^s^−1^)	>1.0/>0.74	>1.47/>2.94	4.7 ± 1/3.6 ± 1 ^4^
***1***/(***K’*_5_*K’*_6_**)	(µM)	135 ± 10 ^5^	34 ± 4 ^5^	3.4 ± 1 ^6^
** *k’* _+5_ **	(s^−1^)	>100 ^7^	>100 ^7^	16 ± 1 ^8^
**P_i_ kinetics of actomyosin**				
** *k’* _+4_ **	(s^−1^)	3.8 ± 1.1	n.d.	14.4 ± 1.0
** *k’* _+4_ *K’* _DPA_ **	(µM^−1^s^−1^)	0.04 ± 0.01	n.d.	0.79 ± 0.08
***1***/***K’*_DPA_**	(µM)	94 ± 32	n.d.	18 ± 3
**Weak-to-strong transition**				
** *k’* ** _weak-strong_	(s^−1^)	1.0 ± 0.01	n.d.	2.6 ± 0.04
**Duty ratio**				
t_strong_/t_total_	%	3 ± 1	<4 ± 1	49 ± 4
Experimental	%	<1	n.d.	35

^1^ Calculated: ***k’*_−A_**/***k’*_+A_**; ^2^ calculated: ***k’*_−DA_**/***k’*_+DA_**; ^3^ calculated from ***k’*_+5_*K’*_5_*K’*_6_**; ^4^ obtained from linear fit of *k*_fast_ of competitive ATP/ADP binding experiment with 25 µM ATP; ^5^ obtained from ADP-inhibition of ATP-induced dissociation; ^6^ obtained from hyperbolic fits of A_slow_ and A_fast_ (Figure 5g); ^7^ estimated form the rate of ADP inhibition of ATP-induced dissociation at excess ADP concentrations; ^8^ y-intercept of hyperbola in Figure 5i; n.d. not determined.

**Table 5 ijms-22-00104-t005:** Parameters used for the estimation of the fractional occupancies of intermediate states.

	Unit	M765^wt^	M765^AA^	M765^GGG^
**Equilibrium constants**				
** *K’* _DPA_ **	(µM^−1^)	0.01	0.01	0.05
** *K’* _4_ **	(µM^−1^)	0.00001	0.00001	0.00001
** *K’* _5_ **	-	0.5	0.74	7.14
** *K’* _6_ **	(µM^−1^)	0.02	0.02	0.02
** *K’* _1_ **	(µM^−1^)	0.0003	0.004	0.001
** *K’* _2_ **	-	8988	1405	624
** *K’* _TA_ **	(µM^−1^)	0.001	0.001	0.001
*K* _3_	-	9.5	9.2	9.6
** *K’* _3_ **	-	95	92	96
**Rate constants of forward reactions**				
** *k’* _+DPA_ **	(µM^−1^s^−1^)	10	10	50
** *k’* _+4_ **	(s^−1^)	3.8	5.0	14.4
** *k’* _+5_ **	(s^−1^)	235	160	20
** *k’* _+6_ **	(s^−1^)	1000	1000	1000
** *k’* _+1_ **	(µM^−1^s^−1^)	100	100	100
** *k’* _+2_ **	(s^−1^)	719	871	923
** *k’* _+TA_ **	(s^−1^)	1000	1000	1000
*k* _+3_	(s^−1^)	38	55	96
** *k’* _+3_ **	(s^−1^)	38	55	96
**Rate constants of backwards reactions**				
** *k’* _−DPA_ **	(s^−1^)	1000	1000	1000
** *k’* _−4_ **	(µM^−1^s^−1^)	0.000038	0.00005	0.000144
** *k’* _−5_ **	(s^−1^)	470	216	2.8
** *k’* _−6_ **	(µM^−1^s^−1^)	20	20	20
** *k’* _−1_ **	(s^−1^)	333,300	250,000	88,500
** *k’* _−2_ **	(s^−1^)	0.08	0.62	1.48
** *k’* _−TA_ **	(µM^−1^s^−1^)	1	1	1
*k* _−3_	(s^−1^)	4	6	10
** *k’* _−3_ **	(s^−1^)	0.4	0.6	1

**Table 6 ijms-22-00104-t006:** Fractional occupancies of actin-attached and actin-detached states of myosin.

	Actin (µM)		ATPase (s^−1^)	Detached ^1^	Weak ^2^	Strong ^3^	Motility (µms^−1^)
**M765^wt^**	0.1K_app_:	9.7	0.30	0.92	0.08	0.003	0.52
	1K_app_:	97	1.67	0.54	0.44	0.020	
	3K_app_:	291	2.49	0.30	0.68	0.020	
	20K_app_:	1940	3.16	0.08	0.89	0.030	
**M765^AA^**	0.1K_app_:	9.7	0.39	0.92	0.08	0.004	0.48
	1K_app_:	97	2.17	0.53	0.45	0.020	
	3K_app_:	291	3.25	0.29	0.68	0.030	
	20K_app_:	1940	4.17	0.07	0.89	0.040	
**M765^GGG^**	0.1K_app_:	1.2	0.66	0.92	0.05	0.030	0.10
	1K_app_:	12	3.73	0.54	0.26	0.200	
	3K_app_:	36	5.64	0.30	0.40	0.300	
	20K_app_:	240	7.17	0.11	0.52	0.390	

^1^ M∙T, M∙D∙P_i_; ^2^ A-M∙T, A∙M∙D∙P_i_; ^3^ A∙M, A∙M∙T, A∙M∙D.

## Data Availability

The data presented in this study are available in this article and also available upon request from the corresponding author.

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
