# Peer review of "Unraveling a Force-Generating Allosteric Pathway of Actomyosin Communication Associated with ADP and Pi Release"

_ijms, 2020, doi:10.3390/ijms22010104_

Round 1
Reviewer 1 Report
The major message of the manuscript is that the W-helix regulates the rate of actin activated ATPase activity through influencing weak to strong actin binding transition and products releases from A.M. The consequence of these kinetics changes is a significantly higher duty ratio and slower sliding in the motility assay. The authors characterized the kinetics of two W-helix mutants and presented the crystal structure of them in ADP.Vi form. The X-ray structure indicates that the mutations did not cause fundamental structural disturbance of myosin, the mutant’s structure is highly similar to that of the wild type. The transient kinetics experiments are presented in a very detailed way and in a very high quality. Also, the experimental design is logic and well-focused. I think this is a very elegant transient kinetics paper, however, I think in some aspects the results are overinterpreted and I may recommend some minor changes, as well.
Step by step questions and comments (major comments are in bold):
- Page 1 line 42-44: I would not use the phrase “primed” for the prehydrolytic step, because this step -called recovery step- is not an irreversible step as priming would implicate but it is an equilibrium of the up and down position states. Furthermore, hydrolysis stabilizes the up lever position state.
- To be consequent with page 1 line 42-44 sentence in Figure 1 I would separate step 3 to 3a and 3b as recovery and hydrolysis steps, respectively.
- Page2 line 59: for the same reason as indicated in comment 1, I would avoid the phrase recovery STROKE, because expression “stroke” indicates an irreversible priming. Thus, I would use recovery step instead.
- Page3 line 92 – the same problem
- Major comment: Page 3 line 114-119: the authors do not explain why specifically T647 and T648 were mutated. They refer an “in depth analysis”, but I think it would be crucial to explain to the readers what that in depth analysis was and why these two aminoacids of W-loop were mutated and why not others? Also, it is not clear why two aminoacids were mutated at the same time, why not single aminoacids and/or step by step.
- Major comment: Also, it would be important to show whether these positions are conserved in the myosin 2 family and in the whole myosin superfamily. It is especially an interesting and important question because even in the myosin2 family the 647 position is mainly N,S or A and very rarely T. Furthermore, there are many myosins in which even 648 position is different from T – however it is more conserved. The results of the paper should be discussed through this aspect as well.
- Major comment: Detailed explanation is required why two aminoacids were replaced by three (3 glicines)
- Major comment: Line 124: labelling the construct as M765AA and M765GGG is misleading, because for a quick reader it may mean as e.g. a M765 motor domain construct with AA or GGG C-terminal extension. It is difficult to find what these constructs are because the constructs are not mentioned in the abstract and it is not easy to find them in the text and/or in the figure legends. I suggest finding more obvious names which directly reflects the mutations.
- Line 182-185: I do not understand your explanation for the reduced motility speed. I think it is pretty obvious that motility slowed dramatically because of the high ratio of strong actin binding state (means 36% duty ratio as showed in Fig 7).
- Fig 7: weak actin binding in Fig7a is misleading because the presented figure indicates that upper cleft binds to actin in weak actin binding state. In contrast, in weak actin binding state predominantly the lower cleft binds to actin and there are only few interactions between the upper one and actin. After closure, actin binding becomes strong mainly because of the new interactions with the upper cleft. I would modify the figure in this way.
- Major comment: Line 446-450. In my opinion there is no direct evidence presented in this paper which confirms this statement. I do not say that I disagree with this statement, but I cannot accept it as a conclusion of the presented results. Please, rephrase it or if you still think that this statement directly comes from your results and your results directly confirms it, please explain it in more detailed step by step way.
- Major comment: Since W-helix is the continuation of loop 2 which is the major actin binding loop of upper cleft, therefore the role of W-helix should be discussed in the context of the role of loop 2. There are several papers (including Acanthamoeba and Chara myosins) that loop 2 variations and phosphorylation regulates actin activated ATPase like mutations in W helix demonstrated in this paper, thus its discussion would be appropriate. Also, I note that loop 2 has a major role in strong actin binding formation and W-helix mutations also affects this step as presented in this manuscript, therefore the authors cannot exclude the possibility that W helix mutation might cause distortion in loop 2 that may cause the demonstrated effect. I could imagine that three glycine in the mutated W helix may dramatically change the dynamics of the W-helix and the connected loop 2 that caused the effect in the actin binding step and consequently the further change in the product release of the actomyosin complex. We cannot exclude that W-loop mutation causes an indirect effect in the appropriate steps through the change in the loop 2 conformation and/or dynamics.
In summary, the manuscript presents a very elegant transient kinetics study with a great implementation. The mutations cause essential and important effect in the actomyosin cycle. Nevertheless, the interpretations of the results should be extended to further possibilities.
Andras Malnasi Csizmadia
Author Response
Please find in the attached file our point-by-point response to the reviews.

Reviewer 2 Report
The authors made and crystallized two mutants of myosin subfragment-1 and, based on the structure and results of detailed measurements of biochemical kinetics and motility assay, explained the mechanism of functional changes due to the alteration in the amino acid residues. The kinetics was thoroughly studied and analyzed properly. I can say the authors fully characterized these mutants.
On the other hand, relation between the structure and function is not completely elucidated. Although it is clear that the modified W-loop affects certain steps of ATPase, precise mechanism is not clarified about how the substitution of two threonines with three glycines interferes interaction between myosin and actin. We should wait for further studies on this point.
I have a few comments on the manuscript.
- Why is the maximum ATP turnover increased while motility is slower? A qualitative explanation will be helpful.
- Figures 1c and 2: It would be helpful to indicate where the replaced threonines are located. It is also helpful to highlight ADP in Figure 2a.
- line 441: Figure 8 is missing.
Author Response
Pleased find in the attached file our point-by-point responses to the reviews.
